# Ongoing repair of migration-coupled DNA damage allows planarian adult stem cells to reach wound sites

Sounak Sahu[1], Divya Sridhar[1], Prasad Abnave[1], Noboyoshi Kosaka[1], Anish Dattani[1], James M Thompson[2], Mark A Hill[2], Aziz Aboobaker[1]*

[1]Department of Zoology, University of Oxford, Oxford, United Kingdom; [2]CRUK/MRC Oxford Institute for Radiation Oncology, ORCRB Roosevelt Drive, University of Oxford, Oxford, United Kingdom

**Abstract** Mechanical stress during cell migration may be a previously unappreciated source of genome instability, but the extent to which this happens in any animal in vivo remains unknown. We consider an in vivo system where the adult stem cells of planarian flatworms are required to migrate to a distal wound site. We observe a relationship between adult stem cell migration and ongoing DNA damage and repair during tissue regeneration. Migrating planarian stem cells undergo changes in nuclear shape and exhibit increased levels of DNA damage. Increased DNA damage levels reduce once stem cells reach the wound site. Stem cells in which DNA damage is induced prior to wounding take longer to initiate migration and migrating stem cell populations are more sensitive to further DNA damage than stationary stem cells. RNAi-mediated knockdown of DNA repair pathway components blocks normal stem cell migration, confirming that active DNA repair pathways are required to allow successful migration to a distal wound site. Together these findings provide evidence that levels of migration-coupled-DNA-damage are significant in adult stem cells and that ongoing migration requires DNA repair mechanisms. Our findings reveal that migration of normal stem cells in vivo represents an unappreciated source of damage, which could be a significant source of mutations in animals during development or during long-term tissue homeostasis.

*For correspondence:
aziz.aboobaker@zoo.ox.ac.uk

Competing interests: The authors declare that no competing interests exist.

## Introduction

Constant threats to genome integrity by both exogenous and endogenous agents has led to the evolution of cellular mechanisms to counteract various forms of DNA damage (*Hoeijmakers, 2001*; *Tubbs and Nussenzweig, 2017*). Accumulated genotoxic damage, particularly in stem cells, is thought to underpin both ageing and the development of cancer (*Vitale et al., 2017*; *Mandal et al., 2011*; *Goodell and Rando, 2015*; *Jackson and Bartek, 2009*). Therefore, maintaining the integrity of the genome is particularly important for longer-lived animals with adult stem cells. We know that efficient DNA repair mechanisms to counteract the effects of replicative stress and other sources of damage are central to avoiding both premature ageing and cancer (*Rossi et al., 2008*; *Jeggo et al., 2016*; *Sperka et al., 2012*; *Adams et al., 2015*). Recent in vitro studies using cancer cell lines, dendritic cells, as well as primary stem cells have shown that migration through micro capillaries or extreme constrictions imparts mechanical stress on nuclei, and this can be a source of DNA damage and genome instability (*Denais et al., 2016*; *Raab et al., 2016*; *Irianto et al., 2017a*; *Smith et al., 2019*; *Nader, 2020*; *Shah et al., 2017*; *Shah, 2020*; *Kirby and Lammerding, 2018*). These studies reveal an assortment of mechanisms by which mechanical stress results in DNA damage. This damage can be either repaired in a regulated manner or cells may undergo differentiation after being reimplanted into animals after in vitro manipulation (*Smith et al., 2019*). However, evidence of

damage caused by purely in vivo adult stem cell migration and whether it is a significant load to DNA repair mechanisms is currently conspicuously absent. If migrating cells, in particular stem cells, experience DNA damage in vivo, this has a number of broad implications for metazoan development and homeostasis. For example, migrating stem cells may experience aberrant effects on stem cell differentiation, become transformed to cause malignancies, or become senescent and contribute to ageing all as a result of DNA damage (*Smith et al., 2019*). As a corollary of this, we would expect that active DNA repair mechanisms might be required to allow continued cell migration in vivo.

However, it is not known whether migration and damage are associated in vivo in normal adult stem cells, as to this point cells have been manipulated to experience migration through artificial constrictions in vitro. To address this, we have improved on existing methods to study DNA damage and repair (DDR) processes in the highly regenerative animal *Schmidtea mediterranea*, focusing on the observation of migrating adult stem cells (*Abnave et al., 2017*; *Guedelhoefer and Sánchez Alvarado, 2012*). We used an established assay in which migrating stem cells can be observed and with which we previously established that stem cells home precisely to wounds, form cell extensions when migrating, and require the transcription factors snail and zeb1, which regulate epithelial to mesenchymal transition, for migration. (*Abnave et al., 2017*; *Guedelhoefer and Sánchez Alvarado, 2012*). Investigating the DNA damage response in this context, we found that the adult stem cell population displays increased levels of DNA damage as they migrate towards a distal wound site and repair this damage when they reach the wound site. We demonstrate that both the repair of DNA damage and active DNA repair mechanisms are required to allow directed stem cell migration to a wound site. Overall, our data find in vivo evidence for the link between cell migration and DNA damage and suggest that a reconsideration of the significance of migratory events across development and adult homeostasis in the context of potential DNA damage is required.

## Results and discussion

### A robust DNA damage response allows stem cells to resist doses up to 15 Gy of ionising radiation

Planarian flatworms have a population of pluripotent adult stem cells, and potentially avoid both ageing and cancer (*Sahu et al., 2017*; *Rink, 2013*; *Wagner et al., 2011*). When exposed to 20 and 30 Gray (Gy) lethal doses of ionising radiation (IR), some smedwi1$^+$ stem cells remain after 24 hr but are not competent to proliferate and rescue the animal. On the other hand, all animals survived exposure to a 15 Gy dose, as amongst the surviving smedwi1$^+$ stem cells some are competent to proliferate and differentiate normally (*Figure 1A,B*, *Figure 1—figure supplement 1A–D*). Stem cell loss after sub-lethal irradiation is dose dependent and continues until 3 days post-irradiation (pi) (*Figure 1B*).

We optimised the use of the COMET assay and staining with antibodies to poly-ADP ribose (PAR), the currently available methods for measuring DNA damage/repair in planarians (*Shibata et al., 2016*; *Wouters et al., 2020*; *Peiris et al., 2016*), to quantify DNA damage in planarian cells. We observe that damage assayed by single-cell gel electrophoresis of whole planarian stem cell populations is IR dose dependent (*Figure 1C,D*), with repair and reduction in COMET signal taking place over the subsequent 11 days (*Figure 1—figure supplement 2A,B*). By combining PAR staining with an antibody to planarian Tudor-1 that marks the perinuclear RNP granules (chromatoid bodies) (*Solana et al., 2009*) in smedwi1$^+$ stem cells (*Figure 1—figure supplement 2C*), we measured damage in stem cells and post-mitotic differentiated cells simultaneously using total nuclear PAR. Levels of PAR staining in the nucleus increased just 5 min after exposure to 5 Gy ionising radiation (IR) (*Figure 1E–G*) and returned to the baseline within 24 hr after exposure (*Figure 1—figure supplement 2D–F*), indicating that this approach accurately measures acute DNA damage events.

We identified conserved DDR genes from the *S. mediterranea* genome and transcriptome (*Grohme et al., 2018*; *Brandl et al., 2016*; *Robb et al., 2008*; *Figure 1—figure supplement 1E*) that are known to be essential elsewhere to repair assorted DNA breaks. The transcripts of most DNA repair genes (*atr*, *atm*, *brca2*, *parp1*, and *parp2*) are enriched in stem cells based on the expression pattern data from sorted stem cells and stem cell progeny and from single-cell expression data (*Dattani et al., 2018*; *Wurtzel et al., 2015*). We found that RNAi of conserved DDR genes

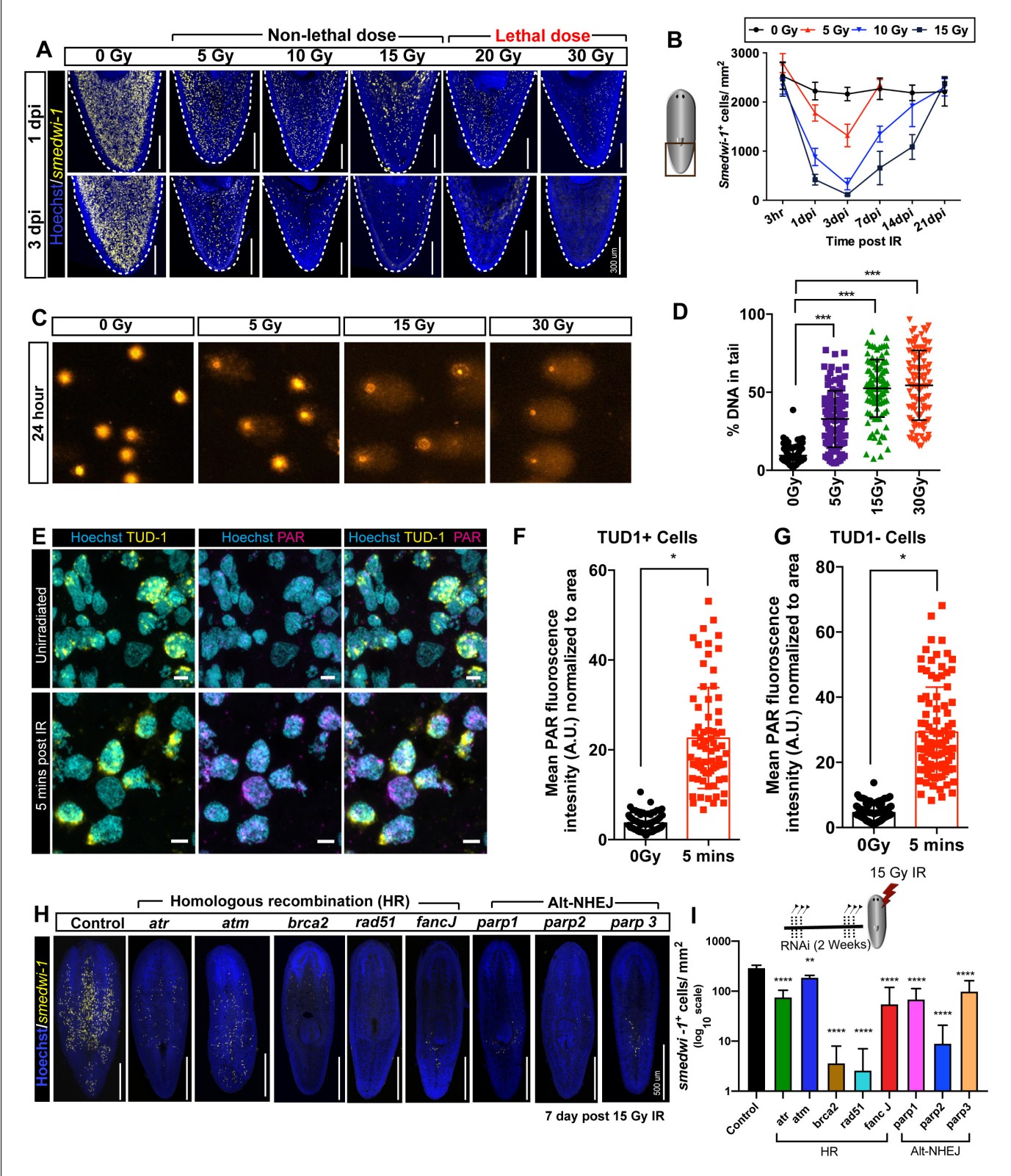

**Figure 1.** Planarian stem cell resistance to doses up to 15 Gy of gamma IR requires conserved DDR pathways. (**A**) smedwi-1 FISH of planarians exposed to different doses of gamma IR (5, 10, 15, 20, and 30 Gy) after 1 and 3 days post-IR (dpi) showing a dose-dependent decrease in stem cell number. Scale bar: 300 μm. (**B**) Quantification of smedwi-1$^+$ cells/mm$^2$ (yellow) showing the repopulation kinetics of surviving stem cells after different doses of IR post-IR (n = 5 per dose, per time point). Results are expressed as mean ± SD. (**C**) COMET assay showing the extent of DNA breaks (comet shape) in

*Figure 1 continued on next page*

*Figure 1 continued*

isolated planarian cells at 24 hr after exposure to 5, 15, and 30 Gy of IR. (D) Quantification of the percentage of tail DNA in COMET assay post-IR at 24 hr. Results are expressed as mean ± SD. Each dot represents the tail DNA in individual planarian cells. (***p<0.0001, one-way ANOVA using Tukey's multiple comparison test). (E) Double immunostaining with Anti-TUD-1 (Yellow) and Anti-PAR (Magenta) showing DNA damage in stem cells (Tud-1$^+$) and post-mitotic differentiated cells (Tud-1$^-$) at 5 min post 5 Gy IR. Nucleus is stained with Hoechst (blue). (F–G) Quantification of PAR fluorescence in Tud1$^+$ and Tud1$^-$ cells normalised to the nuclear area in irradiated and unirradiated cells (0 Gy) (*p<0.0001. Student's t-test). (H) Representative FISH showing stem cell repopulation in Control (gfp) RNAi and after knockdown of different DNA repair genes (involved in homologous recombination [atr, atm, brca2, fancJ, rad51] and Alt-NHEJ [parp1, parp2, parp3]) after 7 days post 15 Gy IR. Gene name represents the RNAi condition. (I) Repopulation of smedwi-1$^+$ cells/mm$^2$ in DDR RNAi worms after 7 days post 15 Gy IR (n = 5 per condition). Results are expressed as mean ± SD in log$_{10}$ scale (***p<0.0001, **p<0.001, one-way ANOVA using Tukey's multiple comparison test).

The online version of this article includes the following source data and figure supplement(s) for figure 1:

**Source data 1.** Numerical data used to make Graphs B, D, F, G, and I.
**Figure supplement 1.** Dynamics of stem cell proliferation and repair kinetics of DNA damage in planarian stem cells.
**Figure supplement 1—source data 1.** Numerical data used to make Graphs B and D.
**Figure supplement 2.** Detection of DNA breaks and DNA damage response in planarian cells after irradiation.
**Figure supplement 2—source data 1.** Numerical data used to make Graphs B, E, and F.
**Figure supplement 3.** Role of DNA repair gene in planarian stem cell maintenance.
**Figure supplement 3—source data 1.** Numerical data used to make Graphs B, D, F, H, J, K, L, M, and N.

individually (*atr*, *atm*, *brca2*, *fancJ*, *parp1*, *parp2*, *parp3*) during normal regeneration and homeostasis did not lead to any phenotypic defects in regenerating animals and did not affect stem cell number or proliferation over a time course of several weeks (*Figure 1—figure supplement 3I–J*). Only RNAi of rad51 led to animal death as previously reported (*Shibata et al., 2016*; *Figure 1—figure supplement 3I–J*). These data suggest that RNAi of these individual genes does not disrupt normal stem cell function during homeostasis. This may reflect the incomplete effects of RNAi, compensation between repair pathways (*Figure 1—figure supplement 3N*), or both.

After sub-lethal IR exposure, surviving stem cells clonally expand to restore homeostatic and regenerative capacity in a dose-dependent manner (*Wagner et al., 2011*; *Wagner et al., 2012*; *Lei et al., 2016*; *Figure 1A,B*, *Figure 1—figure supplement 1A–D*). In this scenario, RNAi of the individual conserved components of homologous recombination (HR) (*atr*, *atm*, *brca2*, *rad51*, and *fancJ*) and alt-non-homologous end-joining pathways (NHEJ) (*parp1*, *parp2*, and *parp3*) after 15 Gy IR led to a failure in stem cell repopulation (*Figure 1H,I*, *Figure 1—figure supplement 3A–H*). As well as establishing robust methods for measuring DNA damage our data provide proof of principle that well-known DDR genes have an ongoing role in stem cell survival, and in DNA repair after IR exposure. This establishes a basis for using *S. mediterranea* as an experimentally tractable in vivo model for studying DDR in adult stem cells. It remains unclear if the lack of phenotypic effect after RNAi of individual DDR genes during normal homeostasis and in the absence of extensive external genotoxic stress is due to incomplete knockdown (*Figure 1—figure supplement 3N*) or compensation between repair mechanisms, or if longer term RNAi experiments over months/years may eventually result in defects with repair machinery has reduced efficiency, but we did not investigate this possibility.

## Migrating stem cells undergo migration-coupled DNA damage that resolves at the wound site

Having established robust DNA damage assays in planarians, we next wished to understand if we could observe whether stem cell migration leads to DNA damage in vivo, as has been observed for mammalian cells constricted in vitro. Planarian stem cells and their progeny must migrate to the site of a wound or during reproductive (asexual) fission to form a regenerative blastema (*Reddien and Sánchez Alvarado, 2004*; *Wenemoser and Reddien, 2010*). Recent work using cells in culture has shown that mechanical stress on the cell nucleus, through a variety of proposed mechanisms, can lead to DNA damage and genome instability during cell migration (*Denais et al., 2016*; *Raab et al., 2016*; *Shah, 2020*; *Lomakin et al., 2020*; *Irianto et al., 2017b*; *Bennett et al., 2017*). However, how important this is generally in vivo in animals is unknown. To study this phenomenon in planarians, despite the lack of live-cell imaging approaches, we established a robust assay for stem cell migration (*Abnave et al., 2017*). This uses a lead shield to perform 'shielded irradiation' and obtain

a stripe of stem cells whose subsequent migration can be followed (*Figure 2—figure supplement 1A–E*). Head amputation triggers anterior migration from the shielded strip of stem cells towards the wound (*Figure 2—figure supplement 1C*) and a lack of posterior cell migration over the experimental time course allows us to clearly define the posterior and therefore the pre-migration anterior boundary of the shield. This system has already allowed the detailed study of planarian stem cell migration in vivo (*Abnave et al., 2017*), for example demonstrating that migrating stem cells develop cytoplasmic projections, precisely home to small 'poke' wound sites and require the activity of transcription factors with conserved roles in epithelial to mesenchymal transition.

Using this assay, we asked if normal stem cell migration in vivo leads to increased DNA damage. Planarian stem cells are characterised by large nuclear-to-cytoplasmic ratios like other animals stem cells (*Aboobaker, 2011*; *Gehrke and Srivastava, 2016*), suggesting that the nucleus in migrating stem cells could encounter physical stress through deformation of normal nuclear shape. In order to check nuclear shape plasticity in migrating cells compare to stationary cells, we measured the nuclear aspect ratio (NAR) (*Chen et al., 2015*) of the cells in the migratory region compared to stationary cells at 7 day post-amputation and observed significant changes in NAR (*Figure 2A–C, Figure 2—figure supplement 1F–G*). Planarian stem cells in the migratory region showed increased NAR (ranging from 1.7 to 2 or higher) (*Figure 2—figure supplement 1F–G*). Consistent with our findings, a recent study using mutant skeletal stem cells also reported that a distribution of NAR ranging from 1.7 to 1.9 induces DNA damage (*Earle et al., 2020*). The magnitude of the change in NAR in migrating planarian stem cells suggested that, by analogy with mammalian stem cells in culture (*Earle et al., 2020*), there was potential for mechanical forces on the nucleus that could cause DNA damage.

We then performed anti-PAR/TUD-1 double immunostaining (*Shibata et al., 2016*), to test whether the changes in nuclear shape in migrating stem cells lead to increased levels of DNA damage. We found that migrating TUD-1$^+$ cells accumulate increased levels of acute DNA damage with increased migratory distance (*Figure 2D,E*, *Figure 2—figure supplement 2B,C*) that eventually return to baseline levels when stem cells reach the wound site, are distributed through the tissue and cease migrating (*Figure 2E*). We note that the migratory regions and wound site have been subjected to the same conditions in the context of shielded irradiation, so damage is not due to the previous exposure of these regions to IR as acute DNA damage, measured by PAR staining, is not observed when migration stops at the wound site.

Stationary stem cells remaining in the shield do not have increased levels of PAR staining, implicating cell migration as the cause of increased DNA damage (*Figure 2D,E*). Similarly, post-mitotic TUD1$^-$ cells (i.e. not stem cells) that are also present in the migratory region and wounded region environment have lower levels of detectable DNA damage than migrating stem cells (*Figure 2F*). This suggests that migrating adult stem cells, which have increased NARs capable of inducing DNA damage, experience ongoing DNA damage events while they migrate and until they reach the wound site and stop migrating.

While PAR staining measures an acute regulatory response to repair new damage, a COMET assay measures global levels of DNA breaks. To further validate that the increased DNA damage observed in stem cells in the migratory region, we performed COMET assays on cells in the shielded and migratory regions of the migration assay (*Figure 2—figure supplement 3A*). We used both intact animals (no migration of stem cells) and animals at 7 days post-amputation (wounded, where stem cell migration is induced). This allowed direct comparison of cell populations from the migratory regions with and without migrating stem cells (*Figure 2G–K*, *Figure 2—figure supplement 3A*). We used in situ hybridisation to *smedwi-1* in animal fragments not used for COMET to confirm the accuracy of separating the shielded and migratory regions (*Figure 2H,J*, *Figure 2—figure supplement 3B–G*). These experiments revealed that the levels of COMET were increased in migratory regions containing stem cells compared to migratory regions from intact animals that are devoid of stem cells (*Figure 2K*). We conclude that migrating stem cells have increased levels of DNA strand breaks while they migrate. These experiments provide an independent measure of DNA damage and also show that migrating stem cells have increased levels of DNA damage.

Overall, these experiments to measure NAR, levels of nuclear PAR, and levels of COMET signal demonstrate planarian stem cells undergo migration-coupled-DNA-damage (MCDD) in vivo. Multiple factors have been shown to lead to increased DNA damage, for example nuclear deformation including changes in the localised concentration of DNA repair factors (*Bennett et al., 2017*) or due

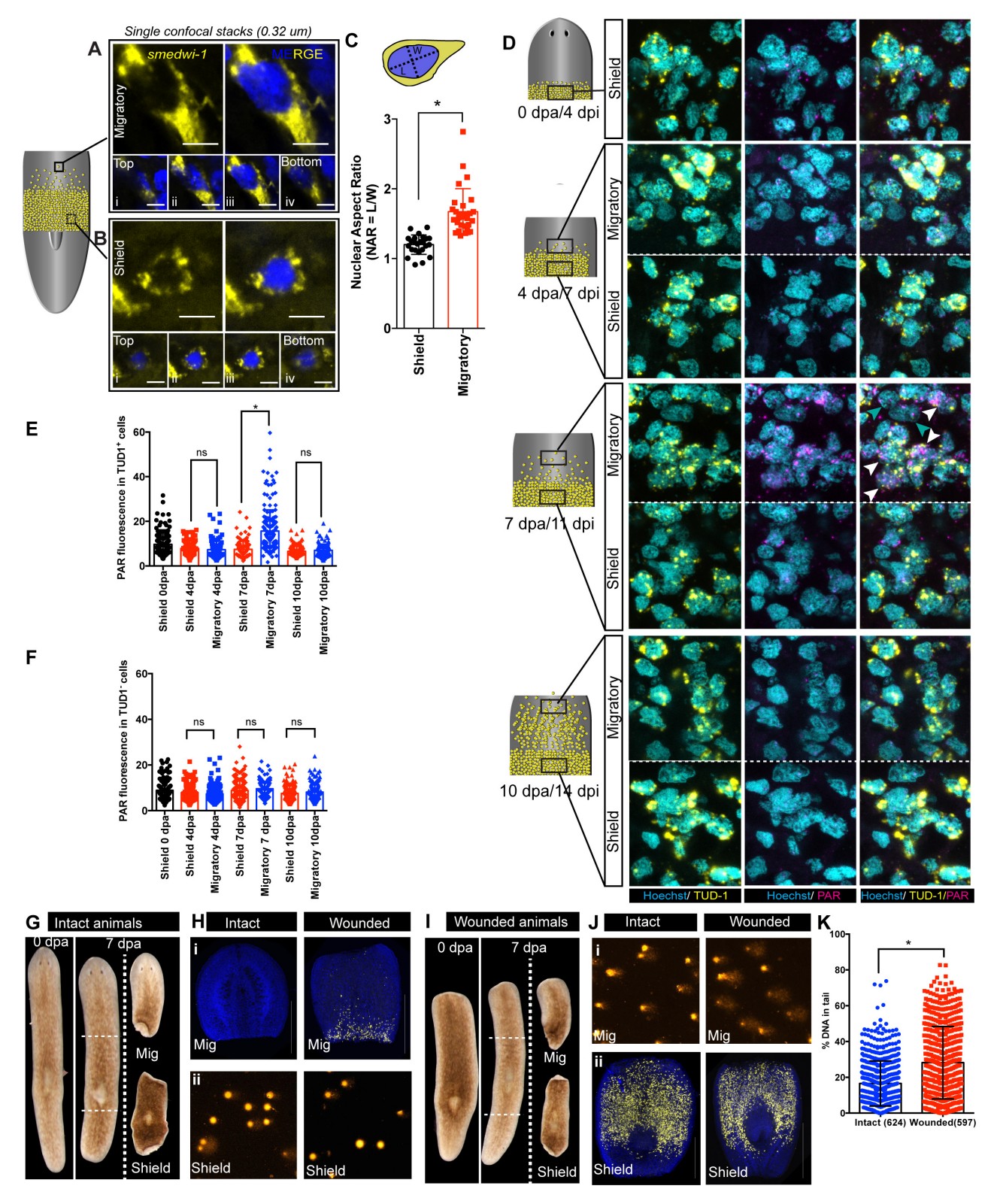

**Figure 2.** Migration-coupled DNA damage (MCDD) in stem cells. Representative FISH showing stem cells (*smedwi-1*[+]) with extended protrusions in migratory cells (**A**) and stationary cells from the shielded region (**B**). Nuclei stained with Hoechst (blue). Images are shown as single confocal Z-stack (0.32 μm). (i–iv) is the single Z-stack from top to bottom. Scale bar: 5 μm. The cartoon explains the setup of shielded irradiation assay where a lead shield is placed in the middle and a lethal dose of 30 Gy is given to these worms. Cells under the shield (Yellow) are protected from IR and starts to

*Figure 2 continued*

migrate after amputation. (C) Quantification of nuclear aspect ratio (NAR) in the migratory cells compared to stationary cells (n = 28 cells in migratory region and n = 24 cells from the shield at 7 dpa/11 dpi [shielded irradiation assay]) (*p<0.0001. Student's t-test). (D) Immunostaining with anti-PAR (magenta) in migrating stem cells (anti-TUD-1, yellow) after 0, 4, 7, and 10 days post-amputation showing MCDD in TUD-1⁺ migrating stem cells. Box denotes the field of cells imaged for analysis. Nuclei stained with Hoechst (cyan). Scale bar: 5 μm. White arrows denote increased nuclear PAR staining in Tud1⁺ stem cells at 7 dpa, and gray arrow denotes lack of PAR fluorescence in Tud1⁻ cells. Quantification showing the PAR fluorescence normalised to the nuclear area from Tud1⁺ stem cells (E) and post-mitotic differentiated Tud1⁻ cells (F) in the migrating region compared to stem cells in the shield. The measurement of PAR fluorescence is strictly nuclear and results are expressed as mean ± SD. (*p<0.0001; one-way ANOVA using Tukey's multiple comparison test). Brightfield images of intact (G) and wounded (I) animals at 0 dpa and 7 dpa showing the amputated migratory region and shielded region. Dotted lines denote the position of the shield. The migratory region was used for *smedwi-1* FISH and the corresponding shielded region was used for COMET assay and vice versa depending on the context (refer to *Figure 2—figure supplement 3*). (H) *smedwi-1* FISH of the migratory tissues at 7 dpa showing the presence of migrating stem cells in wounded animals compared to no migration in intact animals. Cells corresponding to the shielded region were used for COMET assay to check for the extent of DNA damage. (J) *smedwi-1* FISH of the shielded tissue at 7 dpa showing the presence of stem cells under the shield in intact and wounded animals. Cells corresponding to the migratory region from the animals were used for COMET assay to check for the extent of DNA damage in migrating stem cells in wounded animals. (K) Quantification of COMET assay showing the extent of DNA breaks in migrating cells in wounded animals compared to intact animals (absence of migrating stem cells). Each dot represents the percentage of tail DNA from single cells after COMET assay (n = 624 cells from intact animals, and 597 cells in wounded animals). Results are expressed as mean ± SD (student's t-test; *p<0.0001).

The online version of this article includes the following source data and figure supplement(s) for figure 2:

**Source data 1.** Numerical data used to make Graphs C, E, F, and K.
**Figure supplement 1.** Shielded irradiation assay to study stem cell migration and change in nuclear aspect ratio in migrating cells.
**Figure supplement 1—source data 1.** Numerical data used to make Graph E.
**Figure supplement 2.** Migration coupled DNA damage in stem cells.
**Figure supplement 2—source data 1.** Numerical data used to make Graph E.
**Figure supplement 3.** COMET assay to detect DNA breaks during stem cell migration.
**Figure supplement 3—source data 1.** Numerical data used to make Graph E.

to increased replication fork stalling without any disruption of the nuclear envelope (*Shah, 2020*). A reasonable assumption based on our data and in vitro studies suggests that the change in nuclear shape during migration is connected to increased DNA damage in planarian stem cells. While further work requiring the ability to manipulate planarian cells in vitro and perform live imaging will be needed to demonstrate this link more directly, we nonetheless found that stem cell migration in vivo leads to DNA damage in these cells.

## Stem cells pre-loaded with DNA damage incur a delay in migration

We wished to understand whether stem cells in planarians continue to migrate irrespective of DNA damage or whether they repair damage during this process. We hypothesised that stem cells with accumulated DNA damage might pause to remove the source of damage while repairing damage, and then migrate once again. In order to understand, whether the presence of DNA damage acts to inhibit active stem cell migration in vivo, we pre-irradiated whole worms with 5 and 10 Gy IR before following stem cell migratory behaviour in the shielded irradiation assay (*Figure 3A*, *Figure 3—figure supplement 1A*). In these experiments, cells in the shielded region have incurred IR-induced DNA damage. We amputated animals to induce migration within 24 hr to trigger stem cell migration while we knew that DNA damage is still present in these cells (*Figure 1—figure supplement 2B*). We observed that pre-irradiated stem cells undergo a significant delay in migration (*Figure 3B–D*). Although there is a significant delay in migration, stem cells eventually reach the wound site, maintain normal stem cell numbers, and fuel normal regeneration (*Figure 3—figure supplement 1B*). These data suggest a relationship between active migration and levels of DNA damage, with increased levels of damage inhibiting active migration until sufficiently repaired. However, irradiation in this experiment also leads to reduction in cell density and cell proliferation in the shield post-IR exposure, and we cannot exclude that the observed delay in migration may be impart due to these factors (*Pfeifer et al., 2018*).

## Stem cells with MCDD show increased sensitivity to IR

Next, we examined whether stem cells with MCDD are more sensitive to IR. If this were the case, it would demonstrate that the increased damage observed during migration is a significant load on

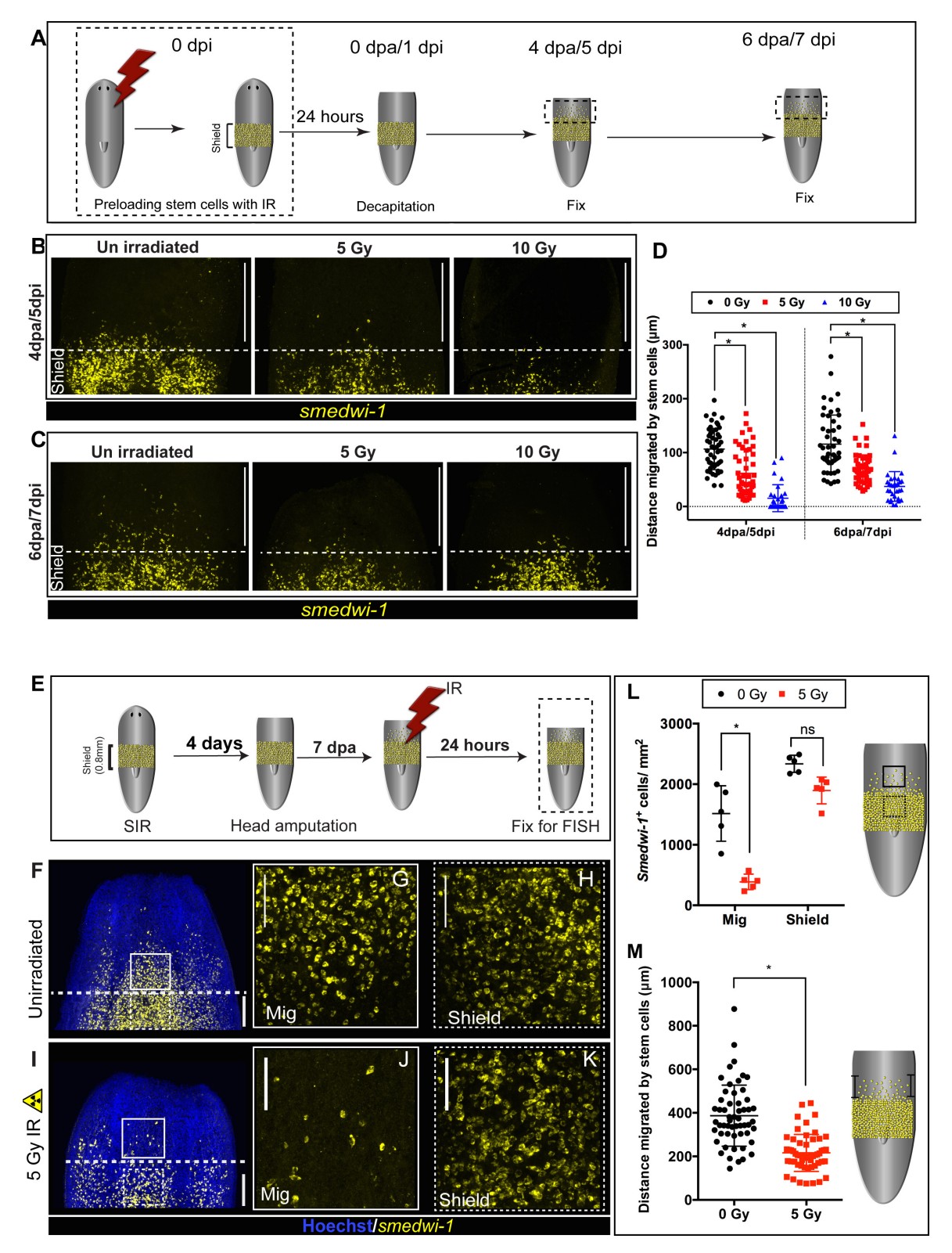

**Figure 3.** DNA damage delays migration and migrating stem cells with MCDD are more sensitive to ionising radiation. (**A**) Experimental scheme showing worms pre-exposed to irradiation (5 and 10 Gy) followed by a shielded irradiation and amputation after 24 hr. Worms are fixed at 4 dpa and 6 dpa (dpa = days post-amputation). Box represents the migratory region, represented in the figure below. (**B, C**) FISH showing worms pre-exposed to IR (5 and 10 Gy) show delayed stem cell migration after 4 and 6 dpa. Dotted line represents anterior boundary of the shield. Scale bar: 350 µm. (**D**)
*Figure 3 continued on next page*

*Figure 3 continued*

Distance migrated by 10 most distant cells are counted from individual worms (n = 5 per condition). Results are expressed as mean ± SD. Statistical significance determined by multiple t-test using the Holm–Sidak method, *p<0.05. (**E**) Schematic of experimental set up to study sensitivity of migrating cells to IR. In addition to the initial shielded irradiation, the worms were irradiated with a low dose of IR (5 Gy, whole body) when MCDD is high (7 dpa) and are fixed after 24 hr to check the survival of the migratory stem cells to IR. (**F–K**) Representative smedwi-1 FISH showing migrating cells are more sensitive to IR than the cells in the shielded region. The region counted for analysis is marked with a box (bold: migratory region, dotted: shielded region). (n = 5 per condition, scale bar: 200 μm F, I; ;100 μm G, H, J, K). (**L**) Quantification of smedwi-1$^+$ cells/mm$^2$ cells in the shielded region and in the migratory region. The decrease in cells/ mm$^2$ in the migratory field is significant compared to the decrease in the shielded region indicating that MCDD sensitises cells to IR. Cartoon showing the region counted for analysis. Each dot represents number of surviving cells from individual worms, n = 5. Statistical significance determined by two-way ANOVA using Tukey's multiple comparison test (*p<0.05). (**M**) Distance migrated by stem cells showing that cells are more sensitive to low-dose IR the further they have migrated. Each dot represents the distance migrated by individual cells. Distance migrated by 11 most distant cells are counted from individual worms (n = 5 per condition). Results are expressed as mean ± SD (student's t-test; *p<0.0001, ns = not significant).

The online version of this article includes the following source data and figure supplement(s) for figure 3:

**Source data 1.** Numerical data used to make Graphs D, L, and M.

**Figure supplement 1.** Stem cells pre-loaded with damage before wounding show delays in migration.

**Figure supplement 1—source data 1.** Numerical data used to make Graphs I and J.

the repair machinery. In addition to performing shielded irradiation, the worms were given an additional dose of 5 Gy to the whole animal at 7 days post-amputation (dpa) (*Figure 3E*), a time point when the highest number stem cells are actively migrating (*Abnave et al., 2017*) and when the cells have highest levels of MCDD as measured by levels of nuclear PAR (*Figure 2E*). We observed a significant decrease in stem cell survival in the migratory region after 5 Gy IR compared to stationary stem cells in the shield (*Figure 3F–M*), demonstrating that migrating stem cells with MCDD are more sensitive to irradiation than stationary cells (*Figure 3H*). We also found that those cells that had migrated furthest, but had not yet reached the wound site, were the most sensitive to IR (*Figure 3M*), suggesting that the accumulation of MCDD may, on average, increase with migratory distance from the wound. We did not observe this striking difference earlier in the migratory process (*Figure 3—figure supplement 1C–J*) when cells have just started migrating in response to wounding, or when cells had already reached the wound site, and migration was complete and MCDD has been repaired (*Figure 3—figure supplement 1K–P*). This demonstrates that increased radiation sensitivity correlates with increased acute levels of DNA damage, dependent on the extent of active stem cell migration (*Figure 2D-E*).

## Wound-induced stem cell migration requires active DNA repair mechanisms to resolve ongoing MCDD

We next asked whether active DNA repair pathways are a functional requirement for continued stem cell migration as they are for recovery from non-lethal doses of IR exposure. We therefore performed RNAi of specific DDR genes in the context of the stem cell migration assay (*Figure 4A*). We observed significantly less migration in *atr*, *atm*, *brca2*, and *parp1* RNAi worms compared to control RNAi worms (*Figure 4B–I*). We did not observe any significant difference in stem cell density in the shielded region (*Figure 4H*), suggesting that knockdown of these genes does not affect normal homeostatic stem cell turnover, as we previously observed (*Figure 1—figure supplement 3I*). The significant reduction in the distance migrated by stem cells (*Figure 4I*) supports a role for active ongoing DNA repair in maintaining genomic integrity during migration and allowing migration to proceed in the face of ongoing damage to the genome. Migrating stem cells incur DNA damage and then either die or differentiate in the context of DDR component knockdowns, similar to the effects of IR exposure. This is supported by the finding that a dose of 5 Gy, which usually removes 40–50% of stationary stem cells after 24 hr (*Figures 1A,B* and *3E–M*), is sufficient to remove over 80% of migrating stem cells.

In future, the development of live-cell imaging and antibodies detecting upstream DNA damage response factors rather than downstream markers of damage foci dependent on repair pathway activity (like PAR staining) will allow us to track cells and observe if stem cells after DDR component RNAi accumulate DNA damage and eventually die due to failure to repair DNA breaks.

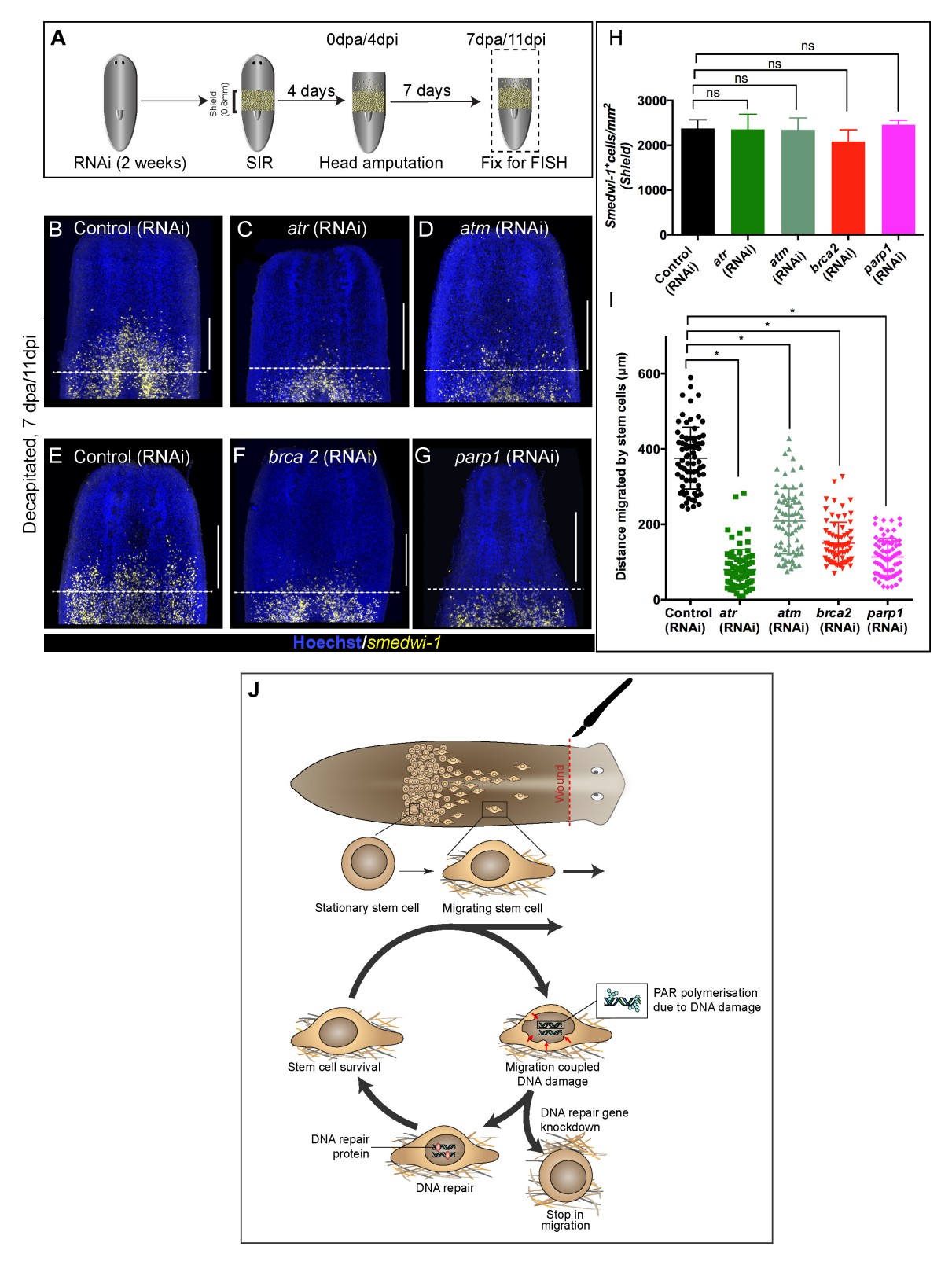

**Figure 4.** Wound-induced stem cell migration requires active DNA repair mechanisms to resolve ongoing MCDD. (**A**) Experimental scheme to study the role of DDR genes in stem cell migration. Worms are injected for 2 weeks (RNAi) followed by the shielded irradiation assay and fixed for FISH 7 days post-head amputation. (**B–G**) Representative smedwi-1 FISH shows migration of stem cells (yellow) at 7 dpa in control (RNAi) (**B and E**) worms, but migration is inhibited in atr (**C**), atm (**D**) brca2 (**F**), and parp1 (**G**) RNAi worms. (n = 5 per RNAi condition). Scale bar: 400 μm, dotted line represents the

*Figure 4 continued on next page*

*Figure 4 continued*

anterior boundary of the shielded region. (**H**) Stem cells in the shielded region show no significant changes in the stem cell turnover. (*p<0.05, ns = not significant, p>0.9999 [*atr*], 0.9818 [*atm*], 0.99997 [*brca2*], 0.3722 [*parp1*], respectively), (n = 5 per RNAi condition, Tukey's multiple comparison test). (**I**) Quantification showing the distance travelled by stem cells after knockdown of DNA repair genes compared to the control RNAi. Each dot represents the distance migrated by individual cells. Distance migrated by 15 most distant cells are counted from individual worms. Results are expressed as mean ± SD n = 75 cells; n = 5 worms/RNAi condition (Tukey's multiple comparison test; *p<0.0001, ns = not significant). (**J**) Stem cells undergo changes in nuclear shape during migration compared to stationary cells in the shield. This model proposes that stem cells undergo migration, followed by MCDD and DNA repair. In the absence of functional DNA repair machinery stem cells fail to migrate.

The online version of this article includes the following source data for figure 4:

**Source data 1.** Numerical data used to make Graphs H and I.

## Conclusions

We have demonstrated that a previously under-appreciated role of DDR is to combat MCDD during stem cell migration in vivo. In the absence of fully functioning DDR machinery, planarian adult stem cells fail to migrate to the wound site in the shielded irradiation assay. Our findings confirm that migration leads to DNA damage in vivo in normal stem cells and, in the light of earlier in vitro findings in cell lines (*Denais et al., 2016*; *Raab et al., 2016*; *Irianto et al., 2017a*; *Nader, 2020*) and mesenchymal stem cells (*Smith et al., 2019*), may represent an evolutionarily conserved cost of this process. An ever growing number of studies using cells in culture or cells constricted in vitro and reimplanted into animals tissues suggest that cell migration mechanically impacts cell nuclei, with effects from complete nuclear rupture at one extreme to moderate nuclear deformation at the other, all leading to DNA damage (*Hoeijmakers, 2001*; *Shah et al., 2017*; *Shah, 2020*; *Kirby and Lammerding, 2018*; *Tubbs and Nussenzweig, 2017*; *Vitale et al., 2017*; *Reddien and Sánchez Alvarado, 2004*; *Wenemoser and Reddien, 2010*; *Lomakin et al., 2020*; *Irianto et al., 2017b*; *Mandal et al., 2011*; *Chen et al., 2015*; *Earle et al., 2020*). We provide evidence of how adult stem cell migration can lead to DNA damage in vivo, thereby providing a physiological relevance to the relationship between cell migration and DNA damage of normal stem cells within a whole organism. Bringing our observations together, we propose a model where migrating cells go through a 'migration–damage–repair–migration' cycle as they move towards the wound site (*Figure 4J*).

Both ageing and oncogenic phenotypes that are commonly thought to be caused by mutations due to replicative stress may also result from genome instability incurred during cell migration. This could be a previously under-appreciated source of further genomic heterogeneity in highly invasive cancer cells that encounter tight spaces in the tissue microenvironment (*Irianto et al., 2017a*; *Irianto et al., 2016*; *Vogelstein et al., 2013*; *Pfeifer et al., 2017*), or a source of mutations in normal homeostasis and development in animals. Future work on naturally occurring MCDD will help to reveal the regulatory interplay between stem cell migration and DNA repair processes.

## Materials and methods

**Key resources table**

| Reagent type (species) or resource | Designation | Source or reference | Identifiers | Additional information |
|---|---|---|---|---|
| Strain, strain background (species) (*Schmidtea mediterranea*) | Asexual clonal line CIW4 | Inhouse laboratory cultured | | All animals used in this study |
| Gene (*Schmidtea mediterranea*) | RAD51 | GenBank | KM487300.1 | RNAi |
| Gene (*Schmidtea mediterranea*) | BRCA2 | GenBank | KT375435.1 | RNAi |

*Continued on next page*

*Continued*

| Reagent type (species) or resource | Designation | Source or reference | Identifiers | Additional information |
|---|---|---|---|---|
| Gene (*Schmidtea mediterranea*) | ATR | Planmine | dd_Smed_v6_8754_0_1 | RNAi |
| Gene (*Schmidtea mediterranea*) | ATM | Planmine | dd_Smed_v6_14586_0_1 | RNAi |
| Gene (*Schmidtea mediterranea*) | FANC-J | Planmine | dd_Smed_v6_16638_0_2 | RNAi |
| Gene (*Schmidtea mediterranea*) | PARP1 | Planmine | dd_Smed_v6_10338_0_1 | RNAi |
| Gene (*Schmidtea mediterranea*) | PARP2 | Planmine | dd_Smed_v6_6154_0_1 | RNAi |
| Gene (*Schmidtea mediterranea*) | PARP3 | Planmine | dd_Smed_v6_2611_0_1 | RNAi |
| Antibody | Anti-digoxigenin-POD, Fab fragments (rabbit polyclonal) | Sigma/Roche | # 11207733910 RRID:AB_514500 | FISH (1:2000) |
| Antibody | Anti-fluorescein-POD, Fab fragments (rabbit polyclonal) | Sigma/Roche | #11426346910 RRID:AB_840257 | FISH (1:2000) |
| Antibody | Anti-H3P (phosphorylated serine 10 on histone H3 (rabbit polyclonal)) | Millipore | #09–797 RRID:AB_1977177 | IF (1:1000) |
| Antibody | Anti-Poly (ADP) Ribose (PAR) monoclonal antibody (clone 10H) (mouse monoclonal) | Santacruz | #SC-56198 RRID:AB_785249 | IF (1:250) |
| Antibody | Anti-TUDOR-1(Tud1) (rabbit polyclonal) | Aboobaker Lab **Solana et al., 2009** | | IF (1:250) |
| Antibody | Goat-Anti-rabbit-HRP (goat polyclonal) | Invitrogen | #65–6120 | IF (1:2000) |
| Antibody | Goat-Anti-mouse HRP (goat polyclonal) | Invitrogen | #62–6520 | IF (1:2000) |
| Chemical compound, drug | Formaldehyde | EMD Millipore | FX0410-5 | Used at 4% for fixing animals |
| Chemical compound, drug | Platinum Taq | Invitrogen | #10966026 | PCR |
| Chemical compound, drug | Trizol | Invitrogen | #15596026 | RNA isolation |
| Chemical compound, drug | Superscript III Reverse transcriptase | Invitrogen | #18080093 | cDNA synthesis |
| Chemical compound, drug | Absolute qPCR mix, SYBR Green | Thermo Fisher | #AB1159A | RT-qPCR expression |
| Chemical compound, drug | Chloretone | Sigma | #112054 | Anaesthetising worms |
| Chemical compound, drug | Instant ocean Sea Salt | Instant Ocean | #SS15-10 | Culturing animals |
| Sequence-based reagent | ATR_F | Sigma | GCGCAGGAATTCAGAAACTC | dsRNA for RNAi |
| Sequence-based reagent | ATR_R | Sigma | GACGGTCACCGAGACCTAAA | dsRNA for RNAi |

*Continued on next page*

*Continued*

| Reagent type (species) or resource | Designation | Source or reference | Identifiers | Additional information |
|---|---|---|---|---|
| Sequence-based reagent | ATM_F | Sigma | ATTCACTGGGCCAACGTTGA | dsRNA for RNAi |
| Sequence-based reagent | ATM_R | Sigma | TCTTCCCTCGACACCAAACG | dsRNA for RNAi |
| Sequence-based reagent | BRCA2_F | Sigma | ATGGACGGGATGTGATGAGC | dsRNA for RNAi |
| Sequence-based reagent | BRCA2_R | Sigma | ATGCACCTTCCACGAGCAAT | dsRNA for RNAi |
| Sequence-based reagent | Rad51_F | Sigma (*Peiris et al., 2016*) | TTTGCAAGGTGGTGTTGAAA | dsRNA for RNAi |
| Sequence-based reagent | Rad51_R | Sigma (*Peiris et al., 2016*) | ATCAGCCAACCGTAACAAGG | dsRNA for RNAi |
| Sequence-based reagent | FancJ_F | Sigma | AGCGGAAAGGAAGACTGTCA | dsRNA for RNAi |
| Sequence-based reagent | FancJ_R | Sigma | TAGGCACGACTTCACTGCAC | dsRNA for RNAi |
| Sequence-based reagent | PARP1_F | Sigma | AACGTGCAATGCTGGAGTTT | dsRNA for RNAi |
| Sequence-based reagent | PARP1_R | Sigma | TCCTACCCCTTTGCAACTGT | dsRNA for RNAi |
| Sequence-based reagent | PARP2_F | Sigma | TGACTGGCAAGATCGTCAGA | dsRNA for RNAi |
| Sequence-based reagent | PARP2_R | Sigma | AGTTGTTCTTGAACCGTGCC | dsRNA for RNAi |
| Sequence-based reagent | PARP3_F | Sigma | AACTCTTGTGGCATGGAACC | dsRNA for RNAi |
| Sequence-based reagent | PARP3_R | Sigma | CGCAGAGTTCGTGAAATGAA | dsRNA for RNAi |
| Sequence-based reagent | Atr_qPCR_F | Sigma | ACGCGTGGTATAGGAGCGTG | qPCR |
| Sequence-based reagent | Atr_qPCR_R | Sigma | TATGACGGTCACCGAGACC | qPCR |
| Sequence-based reagent | Atm_qPCR_F | Sigma (*Peiris et al., 2016*) | CTGATTGGTCGGCTTTCATT | qPCR |
| Sequence-based reagent | Atm_qPCR_R | Sigma (*Peiris et al., 2016*) | AGCTAACCAATCCCCCAAAG | qPCR |
| Sequence-based reagent | Brca2_qPCR_F | Sigma (*Peiris et al., 2016*) | CAAAGAGACCCTGCTTGAGG | qPCR |
| Sequence-based reagent | Brca2_qPCR_R | Sigma (*Peiris et al., 2016*) | AGCCGGAACACAGTACCATC | qPCR |
| Sequence-based reagent | Rad51_qPCR_F | Sigma (*Peiris et al., 2016*) | ATGTCAGAATCCCGATACGC | qPCR |
| Sequence-based reagent | Rad51_qPCR_R | Sigma (*Peiris et al., 2016*) | ATCAGCCAACCGTAACAAGG | qPCR |
| Sequence-based reagent | FancJ_qPCR_F | Sigma | CACCAGTGGAACCTTATCTCC | qPCR |
| Sequence-based reagent | FancJ_qPCR_R | Sigma | GGACGGTCCGTTTCCGATGCT | qPCR |
| Sequence-based reagent | Parp1_qPCR_F | Sigma | CGATTCTATACAATGATGCC | qPCR |
| Sequence-based reagent | Parp1_qPCR_R | Sigma | CTGCTTCCATCAGTTTATAGGC | qPCR |

*Continued on next page*

*Continued*

| Reagent type (species) or resource | Designation | Source or reference | Identifiers | Additional information |
|---|---|---|---|---|
| Sequence-based reagent | Parp2_qPCR_F | Sigma | CAAGAACAACTAATTACGGTGG | qPCR |
| Sequence-based reagent | Parp2_qPCR_R | Sigma | GATCTCGTCGGGTAATATAG | qPCR |
| Sequence-based reagent | Parp3_qPCR_F | Sigma | GATATTGAAAGTACTCAAGC | qPCR |
| Sequence-based reagent | Parp3_qPCR_R | Sigma | CAACATCTAGCATCTTGAACC | qPCR |
| Software algorithm | TBLASTX | U.S. National Library of Medicine | RRID:SCR_011823 | Human gene comparison |
| Software algorithm | BLASTX | U.S. National Library of Medicine | RRID:SCR_001653 | Homology searches |
| Software algorithm | eggNOG.5.0 | European Molecular Biology Laboratory, Hiedelberg | RRID:SCR_002456 | Identify orthologs |
| Software algorithm | Inparanoid | http://inparanoid.sbc.su.se/cgi-bin/index.cgi | RRID:SCR_006801 | Identify orthologs |
| Software algorithm | Planmine | MPI-CBG, Dresden (Dr. Jochen Rink) | http://planmine.mpi-cbg.de/ | Identify flatworm sequences |
| Software algorithm | Fiji/Image-J | MPI-CBG, Dresden/ National Institutes of Health (NIH) | PMID:22743772 RRID:SCR_002285 | Image processing and analysis |
| Software algorithm | KOMET (andor) | Oxford instruments | https://andor.oxinst.com/products/komet-software/ | Comet assay analysis |
| Software algorithm | Graphpad Prism v6 | Graphpad | RRID:SCR_002798 | Graphs and statistical analysis |
| Software algorithm | Illustrator CC | Adobe | RRID:SCR_010279 | Making figures |

## Planarian culture

Asexual freshwater planarians of the species *S. mediterranea* were used in this study. The culture was maintained in 0.5% instant ocean solution (which we have referred to as planarian water in this paper) and fed with organic calf liver twice a week. Planarians were starved for 7 days prior to each experiment and also throughout the duration of each experiment and cultured in the dark at 20℃.

## Gene cloning and RNAi

The sequences of *S. mediterranea* RAD51 and BRCA2 were described previously (*Peiris et al., 2016*). Planarian DDR genes were identified by BLAST searches against the Planmine database, leading to the identification of full-length mRNA transcripts (details in key resources table). Fragments of these genes were cloned into the pPR-T4P plasmid vector containing opposable T7 promoters (kind gift from Jochen Rink, MPI Dresden). These clones were used for in vitro transcription to synthesise dsRNA and RNA probes as previously described in *Abnave et al., 2017*. dsRNA was delivered via microinjection using Nanoject II apparatus (Drummond Scientific) with 3.5'' Drummond Scientific (Harvard Apparatus) glass capillaries pulled into fine needles on a Flaming/Brown Micropipette Puller (Patterson Scientific). Worms were injected with $3 \times 32$ nl of dsRNA six times over 2 weeks. A 1 day gap was kept between the last injection and irradiation experiments (as described in *Abnave et al., 2017*). The primers used for amplification of DNA for dsRNA synthesis/RNA probes can be found in key resources table. Identification of orthologous genes across animal species was done using the Inparanoid database (*O'Brien et al., 2005*) (http://inparanoid.sbc.su.se/cgi-bin/index.cgi) and EggNOG database (*Huerta-Cepas et al., 2019*) (http://eggnogdb.embl.de). The phylogenetic tree is based on that presented by *Grohme et al., 2018*. We also used reciprocal blastp result against the nr database and tblastn result against each sequence. The Planmine database

(*Rozanski et al., 2019*) was used for the identification of sequences of *S. mediterranea* and other flatworm species.

## Gamma irradiation

Animals were starved for at least 7 days and exposed to 1.5, 5, 10, 15, 20, and 30 Gy of 137 Cs gamma rays (for *Figure 1A*) using a GSR D1 Gsm (Gamma-Service Medical GmbH, Leipzig, Germany) gamma irradiator at a dose rate of 1.9 Gy/min. This device was also used to apply doses to whole worms before or after shielded irradiation.

## Shielded irradiation assay

Shielded irradiation was performed as previously described (*Abnave et al., 2017*). Worms (3–5 mm) were anesthetised in ice-cold 0.2% chloretone and aligned on 60 mm Petri dish. The Petri dish was pre-marked with a line at the bottom according to the dimension of the shield (as described in *Figure 2—figure supplement 1A*). The anterior tips of individual worms were aligned to keep the absolute migratory distance between the tip of head and the shielded region fixed. The Petri dish containing worms was then placed directly on top of the lead shield and irradiated from below with X-rays (225 kV, 0.5 mm Al filter, 23 Gy/min). The head and tail regions of the worms received 30 Gy, while the shielded region received less than 1.5 Gy (*Figure 2—figure supplement 1B*). Immediately following irradiation, the worms were placed into fresh planarian water and cultured in the dark at 20°C. For experiments involving an initial dose of gamma irradiation, worms were incubated for 15 min in planarian water before being used for shielded irradiation assay (Experiment in *Figure 3A*). Heads were amputated 4 days post-shielded irradiation to induce migration towards the wound (considered as 0 day post-amputation [dpa] or modified as necessary for a particular experiment [e.g. *Figure 3A*]). Lack of posterior cell migration in the absence of a posterior wounds allowed us to define the boundary of the shield and measure the distance migrated by stem cells in whole mount samples as previously described (*Abnave et al., 2017*).

## COMET assay in planarians

Frosted microscope slides were coated with 700 µl of 1% normal-melting-point agarose (NMPA) in 1× PBS to make a uniform layer and dried overnight at 55°C. Worm fragments were gently diced to minimise any mechanical stress to the cells. The tissue pieces were digested using Papain (15 U/ml) for 1 hr at 25°C. Pieces were mechanically dissociated using a P1000 pipette to form a single-cell suspension and filtered through 100 µm and then 35 µm cell strainers (BD Falcon). Ten thousand dissociated single cells were re-suspended in 80 µl of $CMFHE^{2+}$, and an equal amount of 1.5% low-melting-point agarose was added and mixed. Forty microlitres of the cell-agarose suspension was added onto an NMPA-coated slide and allowed to solidify at 4°C. Slides were incubated overnight (~15 hr) in a coplin jar at 4°C with lysing solution ([2.5 M NaCl, 100 mM EDTA, 10 mM trizma base, NaOH added to pH 10.0] and freshly added 1% triton x-100). This solution was then replaced with a neutralisation buffer (0.4 M Tris base in $dH_2O$, pH to 7.5) for 15 min at 4°C. Following this, the neutralisation buffer was then removed, and the slides were placed into an electrophoresis chamber at 4°C filled with freshly prepared 1× electrophoresis buffer (300 mM NaOH and 1 mM EDTA in $dH_2O$) at 4°C. The slides were allowed to equilibrate for 15 min followed by an alkaline electrophoresis at 20 V for 20 min at 4°C. Next, slides were transferred back into the coplin jar and equilibrated for 5 min in neutralisation buffer. The slides were stained with SYBR Green I (1:10,000 dilution) in freshly prepared 1× TE buffer (10 mM Tris–HCl and 1 mM EDTA, pH 7.5). For long-term storage, the slides were fixed with cold 100% ethanol for 5 min and dried. After drying, 50 comets per slides were analysed using KOMET software (Andor), and the percentage of tail DNA was measured after different doses of gamma irradiation or from 'shielded' regions or 'migrating' regions both with and without migrating stem cells (i.e. with or without wounding). The shielded and unshielded areas were amputated away from each other based on visual estimation. In order to estimate the accuracy of the dissected areas, we performed Comet assay or *smedwi*-FISH on the same animals, such that if a piece (shielded) is used for Comet assay, then the corresponding part (unshielded migratory) is used for *smedwi-1* FISH to assure the accuracy of amputation.

Each FISH was performed in three replicates (InM1, InM2, InM3) with three worms per batch ('In' or 'Wo' corresponding to Intact or Wounded. 'M' and 'S' corresponds to Migratory tissue or

Shielded tissue). The corresponding tissues from these worms were pooled by experimental groups for use in the COMET assay. Individual replicates were analyzed for the presence of *smedwi-1* cells to confirm the accuracy of separating shielded vs unshielded areas.

For comet assay, cells were embedded into four to six slides/replicate, and 50 comets were randomly scored from each slide to a total of 200–300 comets analyzed per condition per replicate. Each of these replicates are shown in *Figure 2—figure supplement 3A–G*, with the detailed schematic on the experimental plan and subsequent analysis.

The extensive DNA fragmentation that occurs during apoptosis does not show comet-like structures; hence, comet tails are representative of DNA breaks in remaining cells. Fragmented DNA in an apoptotic nuclei is very small (size of a nucleosome oligomer) and would generally disappears completely by diffusion in the gel during lysis and/or electrophoresis and are mostly seen as diffuse spheroid or like a halo around the nucleoid-head in comet images and does not show a structure that resembles comet-tail. (For interpretation of Comet assay results, see *Lorenzo et al., 2013*; *Collins, 2004*.)

## In situ hybridisation and immunostaining

All fluorescence in situ hybridisation (FISH) experiments are performed against a particular mRNA of interest, thereby allow detection of the mRNA expression. FISH was performed as described in *King and Newmark, 2013* and previously reported sequences were used for riboprobe synthesis of smedwi-1 (*Abnave et al., 2017*). The antibodies used for immunostaining were anti-H3P (phosphorylated serine 10 on histone H3; Millipore; 09–797; 1:1000 dilution *Abnave et al., 2017*), anti-poly (ADP) ribose (*Shibata et al., 2016*) (PAR) monoclonal antibody (1:250) (Santacruz, clone 10H), and anti-TUD1 (1:250 dilution, based on *Solana et al., 2009*). Anti-rabbit-HRP (H3P and TUD-1) and anti-mouse-HRP (PAR) (1:2000 dilution) secondary antibodies were used followed by tyramide signal amplification for FISH and immunostaining as described in *King and Newmark, 2013*.

## Sectioning of planarian worms

Planarians were killed in 2% HCl and Holtfreter's solution and fixed with 4% formaldehyde for 2 hr. The worms were then washed in PBSTx (0.3% triton-X) and dehydrated with an increasing gradient of methanol washes and stored at −20°C. The following day worms were re-hydrated with a decreasing gradient of methanol and PBS, Xylene washes (two washes of 7 min each) and placed in molten paraffin for 1 hr. Individual worms were then aligned (sagittal or transverse) in paraffin moulds, trimmed, and sliced into 10 µm sections using a microtome. Individual ribbons of planarian sections were placed in a 37°C water bath and aligned to have the entire worm on each poly-lysine-coated slide.

## Immunostaining on paraffinised sections

Planarian sections were deparaffinised using xylene substitute (two washes of 7 min each) and washed with PBS-Tx0.3 (0.3% triton-X). The sections were subjected to antigen retrieval with Trilogy (Cell Marque) at 90°C. The slides were then fixed in 4% formaldehyde for 15 min followed by two washes of PBS-Tx0.5 (0.5% triton-X) for 30 min and transferred to a blocking solution (0.5% BSA and PBS-Tx0.5). One hundred and fifty microlitres of primary antibody (diluted in blocking solution) was added to individual slides, and a parafilm was placed on top for uniform spreading of the antibody solution. After an overnight antibody incubation at 4°C, the slides were washed with alternating changes of PBS-Tx0.5 and PBS + 0.1% tween-20. The secondary antibodies were diluted in blocking solution and incubated overnight. The slides were washed again with alternating changes of PBS-Tx0.5 and PBS + 0.1% tween-20. After two 10 min washes, slides were developed with Tyramide/other fluorophores. For double immunostaining, sodium-azide based peroxide inactivation was performed after the development of each antibody. After two 10 min washes with PBS-TW, slides were stained with Hoechst for nuclear staining overnight. Slides were mounted and imaged using a 100× oil objective lens in an Olympus FV1000 confocal microscope with the appropriate fluorescent lasers.

## Image processing and data analysis

Whole worm confocal imaging was done with Olympus FV1000 and taken as Z-stacks (slices of 4 µm each D/V axis) that were stitched and then processed as a maximum projection using Fiji software (https://fiji.sc/). All measurements and quantifications were done with Fiji (using cell counter plug in) and normalised to the area. Images in *Figure 2A,B, i-iv* are single confocal stacks (0.32 µm) taken with a Zeiss 880 Airyscan microscope using a 63× oil objective lens and manually cropped into individual cells for counting in Fiji software. Nuclear aspect ratio was measured taking the ratio of the length and the width of the nucleus (*Chen et al., 2015*). Images were then processed and cropped before pseudo-colouring the signals in Fiji. The background is set to black for better visualisation, and all figures are prepared using Adobe Illustrator v6 with colour combinations in CMYK format to make scientific figures accessible to people with colour-blindness. Quantification of PAR fluorescence was performed in Fiji using the 'mean fluorescence analysis' tool and normalised to the nuclear area using the Hoechst signal. Total PAR fluorescence was measured from all the nuclei, with high intensity of perinuclear TUD1 staining used to determine Tudor-1-positive stem cells (*Solana et al., 2009*).

## Quantitative RT-qPCR

Total RNA from samples of three to five worms were extracted with Trizol reagent (Invitrogen) according to the manufacturer's instructions for each of two biological replicates for each RNAi condition. RNA was treated with TURBO DNase (Ambion). First-strand cDNAs were synthesised with SuperScript III reverse transcriptase (Invitrogen) and qRT-PCR experiments used the Absolute qPCR SYBR Green Master Mix (Thermo Scientific). Experiments were performed on two biological replicates per RNAi condition. Each biological replicates was technically replicated three times, with each technical replicate consisting of three replicate amplification reactions. Primers for DDR gene are listed in the key resources table. *Smed-ef-2* was used for normalisation using primers described previously (*Solana et al., 2013*).

## Statistical analysis

Results are expressed as mean ± standard deviation (SD). Statistical analyses were performed using GraphPad Prism version 6.0 (https://www.graphpad.com/). Student's t-test and Tukey's multiple comparison test were used for statistical significance at $p < 0.05$. Exact p-values are reported for all experiments.

## Acknowledgements

This work was supported by the Medical Research Council (Grant numbers MR/M000133/1, MR/T028165/1), Biotechnology and Biological Sciences Research Council (Grant number BB/K007564/1) awarded to AAA. SS is funded by the Clarendon Scholarship (University of Oxford) and by the Elizabeth Hannah Jenkinson Fund. DS is funded by the Oxford-Merton-Natural Motion Graduate Scholarship. NK was funded by a Marie Sklodowska Curie individual fellowship by Horizon 2020. AD is funded by a BBSRC DTP studentship (BB/J014427/1). MAH and JMT acknowledge funding from the MRC Strategic Partnership Funding (MC-PC-12004) for the CRUK/MRC Oxford Institute for Radiation Oncology.

## Additional information

### Funding

| Funder | Grant reference number | Author |
| --- | --- | --- |
| Medical Research Council | MR/M000133/1 | Aziz Aboobaker |
| Biotechnology and Biological Sciences Research Council | BB/K007564/1 | Aziz Aboobaker |
| University of Oxford | Clarendon Scholarship | Sounak Sahu |
| University of Oxford | Natural Motion Scholarship | Divya Sridhar |

| | | |
|---|---|---|
| H2020 Marie Skłodowska-Curie Actions | | Noboyoshi Kosaka |
| Biotechnology and Biological Sciences Research Council | BB/J014427/1 | Anish Dattani |
| Medical Research Council | MC-PC-12004 | James M Thompson |
| Medical Research Council | MR/T028165/1 | Aziz Aboobaker |
| University of Oxford | Elizabeth Hannah Jenkinson Research Fund | Sounak Sahu |

The funders had no role in study design, data collection and interpretation, or the decision to submit the work for publication.

### Author contributions
Sounak Sahu, Conceptualization, Data curation, Formal analysis, Funding acquisition, Investigation, Visualization, Methodology, Writing - original draft, Writing - review and editing; Divya Sridhar, Anish Dattani, Investigation; Prasad Abnave, Conceptualization, Formal analysis, Supervision, Investigation, Visualization, Methodology; Noboyoshi Kosaka, Supervision, Investigation, Methodology; James M Thompson, Formal analysis, Supervision, Investigation, Methodology; Mark A Hill, Supervision, Methodology, Project administration; Aziz Aboobaker, Conceptualization, Formal analysis, Supervision, Funding acquisition, Investigation, Methodology, Writing - original draft, Project administration, Writing - review and editing

### Author ORCIDs
Aziz Aboobaker (iD) https://orcid.org/0000-0003-4902-5797

### Decision letter and Author response
Decision letter https://doi.org/10.7554/eLife.63779.sa1
Author response https://doi.org/10.7554/eLife.63779.sa2

## Additional files

### Supplementary files
• Transparent reporting form

### Data availability
All data generated or analysed during this study are included in the manuscript and supporting files.

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
