## [Decision Letter]

**Acceptance summary:**

We would like to congratulate the authors for their work focused on studying responses to wounding in planarians. They find that upon planaria decapitation, stem cell migration to the wound site is accompanied by DNA damage occurring in the migrating stem cells. These findings are obtained by using a combination of techniques, which include RNA interference, comet assays, PAR immunohistochemistry, and a shielded radiation assay. Their findings echo findings in vertebrates, where cell migration in vitro can be a source of DNA damage. Hence, migration-dependent DNA damage seems to be an ancient mechanism, shared between flatworms and vertebrates. The authors show that migrating stem cells are susceptible to irradiation-induced DNA damage, which delays migration. Additionally, the authors show that successful migration to the wound site requires a substet of DNA-repair genes.

**Decision letter after peer review:**

Thank you for submitting your article "Ongoing repair of migration-coupled DNA damage allows planarian adult stem cells to reach wound sites" for consideration by *eLife*. Your article has been reviewed by 3 peer reviewers, including Dario Riccardo Valenzano as the Reviewing Editor and Reviewer #1, and the evaluation has been overseen by Marianne Bronner as the Senior Editor.

The reviewers have discussed the reviews with one another and the Reviewing Editor has drafted this decision to help you prepare a revised submission.

Summary:

Sahu et al. investigate a specific class of responses to wounding in planarians. Specifically, they find that upon decapitation, stem cell migration to the wound site is accompanied by DNA damage occurring in the migrating stem cells. These findings echo findings in vertebrates, where cell migration in vitro can be a source of DNA damage. Hence, migration-dependent DNA damage seems to be an ancient mechanism, shared between flatworms and vertebrates (although so far shown only in vitro). The authors show that migrating stem cells are susceptible to irradiation-induced DNA damage, which delays migration. Additionally, the authors show that successful migration to the wound site requires a substet of DNA-repair genes.

While we feel this work has a great potential, the major conclusion that links DNA damage in stem cells to migration is not justified yet and requires additional experiments to support this claim.

Could migrating cells be more sensitive to oxidative stress when entering the wounded region?

Could DNA damage be coupled with a specific phase of the cell cycle?

The paper needs thorough proofreading (e.g. "However, it is not known to whether migration and damage are associated in vivo in normal adult stem cells, as to this point cells have studies have been manipulated to experience migration through constrictions in vivo").

Essential revisions:

Before being considered for publication in *eLife*, the authors need address

the following outstanding issues:

1. Disentangle migration from cell proliferation. One caveat to the interpretation that stem cells are migrating is that healthy cells from the shielded area, having suffered no DNA damage, are capable of dividing, and produce new daughter cells that eventually repopulate the animal. To strengthen support for stem cell migration, the cell cycle should be inhibited with nocodazole after shielded radiation (as used in Grohme, Nature 2018, van Wolfswinkel, Cell Stem Cell 2014). If cells are indeed migrating and not proliferating, equivalent numbers of cells should be present outside of the shielded area after nocodazole treatment. If fewer, this will suggest that the apparent migration arises from proliferating cells moving beyond the shielded region via expansion of this population rather than directed migration. Alternatively, to clarify whether the wound's biochemical environment – without a regenerative response that requires cell migration – could trigger DNA damage in stem cells, the authors could check if wounding in the shielded region induces DNA damage in non-migrating stem cells. Can the authors assess whether migrating cells are more sensitive to oxidative stress when entering the wounded region, or whether DNA damage could be coupled to a specific phase of the cell cycle?

2. Can degradation due to apoptosis be compatible with the comet assay results?

In Figure 3, can longer comet tails be a result of increased apoptosis after radiation exposure?

3. RNAi results are incompletely reported. According to the paragraph in the middle of page 4, data for knockdown of atr, atm, brca2, parp1, and parp2 under homeostatic conditions should be included in Figure S3. However, the data in this figure is phosphohistone-H3 staining instead of smedwi-1 staining, used in most other figures. In addition, the data in the figure are incomplete (not representing all genes tested). Rad51 is only mentioned, not shown. Approximate numbers of smedwi-1 cells in homeostatic conditions (either by qRT-PCR or in situ hybridization) need to be shown. Controls showing the efficacy of RNAi need to be included (either by qRT-PCR or in situ hybridization). The authors mention multiple times that they may be getting incomplete knockdown – why not just include some validation? Furthermore, why is ATM RNAi excluded from Figure 1H (and methods?).

4. If the inability to repair DNA is solely responsible for a failure of stem cells to migrate to the wound, then cells should be seen dying en route to the wound, with elevated levels of PAR staining and increased comet tails. Is this the case? The authors need to verify PAR staining specificity. An important control for these experiments would be a whole-animal stain of a shielded animal, which should show that the shielded area is devoid of PAR staining. The authors also need to indicate that quantification of PAR fluorescence was done in a standardized way, some indication of the number of animals analyzed, etc. Furthermore, the color selection makes it very difficult to see the staining in some images (e.g. Figure 2D). Higher magnification images and single-color images should be included. How was the presence/absence of DDR components assessed?

5. Cite Guedelhoefer, Development, 2012, as this was the first publication to suggest that stem cell migration occurs in planarians.

6. What is the relationship between the distance travelled by the stem cells and the amount of DNA damage? The authors already have the images at 4dpa/7dpi to extract this data.

---

## [Author Response]

[…] While we feel this work has a great potential, the major conclusion that links DNA damage in stem cells to migration is not justified yet and requires additional experiments to support this claim.Could migrating cells be more sensitive to oxidative stress when entering the wounded region?

We thank the reviewers for the suggestion to further clarify if migrating cells are more sensitive to oxidative stress arising from the wound’s biochemical environment. Given cells arriving at the wound site show reduced damage compared to migrating cells we do not believe this to be an issue. In addition, immunostaining with anti-PAR (to detect DNA damage) and Anti-Tud1 (to detect damage in stem cells (Tud1+) and differentiated progenies (Tud1- cells)) allowed us to perform a controlled experiment where we can simultaneously analyse both migratory and non-migratory cells in the same environment. The non-migratory Tud1- differentiated cells that are present in the migratory region and at the wound site region presumably would also be more sensitive/ acquire DNA damage due to “oxidative stress” as the reviewers suggested. However, we only see increased DNA damage in TUD1+ stem cells in the migratory region compared to the shield and do not detect increased damage in TUD1- cells (Figure 2 D-F). We have also confirmed this result by performing COMET assay using cells from the shielded region and migratory regions with and without migrating stem cells (Figure 2 G-K). We don’t think cells in the migratory region or at the wound site are acquiring DNA damage because of oxidative stress from the wounded region.

Could DNA damage be coupled with a specific phase of the cell cycle?

The activation of DNA damage checkpoints is accompanied by cell-cycle arrest, that provides a temporal delay to accommodate DNA repair factors to repair DNA lesions before resuming cell proliferation. NHEJ based DNA repair is effective throughout all the phases of cell cycle, whereas HDR requires a homologous template to repair the double stranded break, it occurs during G2/M phase of the cell cycle. It could be that damage is repaired through the cell cycle as migrating cells stop and divide and we have not investigated this in this study and do not have the resources available to do this in a reasonable time frame.

The paper needs thorough proofreading (e.g. "However, it is not known to whether migration and damage are associated in vivo in normal adult stem cells, as to this point cells have studies have been manipulated to experience migration through constrictions in vivo").

We thank the reviewer for pointing this out. We have proofread the manuscript and fixed the mistakes in some sentences where we could see them. These changes are visible as tracked changes in the manuscript.

Essential revisions:Before being considered for publication in eLife, the authors need addressthe following outstanding issues:1. Disentangle migration from cell proliferation.

Firstly, we do not think these processes can be disentangled in a proliferating cell population such as the planarian neoblasts, they are likely to be inextricably linked as neoblasts are always proliferating. So we are not sure if this is possible.

One caveat to the interpretation that stem cells are migrating is that healthy cells from the shielded area, having suffered no DNA damage, are capable of dividing, and produce new daughter cells that eventually repopulate the animal.

This is correct as animals placed through the assay eventually regenerate normally to form healthy animals, this was reported in detail in our previous work where the migration assay was established (Abnave et al. 2017, Development). So, this is not a caveat. We have added a little more detail about this previous work to the manuscript.

To strengthen support for stem cell migration, the cell cycle should be inhibited with nocodazole after shielded radiation (as used in Grohme, Nature 2018, van Wolfswinkel, Cell Stem Cell 2014). If cells are indeed migrating and not proliferating, equivalent numbers of cells should be present outside of the shielded area after nocodazole treatment.

We have shown in Abnave et al., 2017 Development that stem cells migrate, form extended cell membrane processes as they migrate and fail to migrate towards the wound after knockdown of EMT transcription factors, a matrix metalloprotease, a beta-integrin etc without any effect on proliferation. We also reported the dynamics of proliferation of the migrating population and their ability to produce post-mitotic progeny of the epidermal lineage.

If fewer, this will suggest that the apparent migration arises from proliferating cells moving beyond the shielded region via expansion of this population rather than directed migration.

In our previous study we clearly showed accurate homing of cells directly to the site of small wounds caused by fine needles, with considerable accuracy. Cells did not migrate Given the speed, spread and spacing (and functional work) we are convinced (as were reviewers at the time) that this is bona-fide migration of stem cells, not spreading through proliferation (see Abnave et al., 2017 Development).

We thank the reviewer for the thoughtful suggestion for using Nocodazole to stop cell cycle progression and check for migration. We are concerned about the specificity of any result after Nocodazole treatment as while it blocks the cell cycle as shown in Grohme et al. 2018 and Van Wolfswinkel et al. 2014, it exerts its effect by interfering with microtubule polymerization. Cytoskeleton disassembly plays a major role in cell migration, and Nocodazole treatment has a significant impact on directed cell migration as observed in migration of cells in culture for example (Ganguly et al., JBC 2012; PMID-23135278; Baudoin et al., 2008 Dev Neurosci; PMID-18075261). Hence, we do not think that this experiment helps us beyond the extensive previous characterization of Abnave et al. 2017.

It is correct that cells are dividing and differentiating as they migrate. Cell proliferation and differentiation (without migration) also happens in the shield where we detect no rise in DNA damage. Similarly, we also don’t see a rise in DNA damage in non-migrating TUD1^-^ post-mitotic differentiated cells (Figure 2 E-F). This data strongly supports our hypothesis that migration leads to DNA damage in Tud1^+^ stem cells.

Moreover, the inability of *smedwi-1+* stem cells to migrate after RNAi of DDR genes (Figure 4A) prove that cells are dying/differentiating en route to the wound, while cells in the shield are maintained. This further disentangles the role of migration and proliferation because we don’t see any significant change in the number of *smedwi-1+* cells in the shielded region, suggesting healthy stem cells starts to migrate to the wound and acquire damage during migration and eventually die or differentiate due to the failure to resolve increased DNA damage. Together with multiple lines of evidence our data strongly confirms our interpretation that indeed migration is causing the DNA damage and not aberrant cell-proliferation.

Alternatively, to clarify whether the wound's biochemical environment – without a regenerative response that requires cell migration – could trigger DNA damage in stem cells, the authors could check if wounding in the shielded region induces DNA damage in non-migrating stem cells.

We thank the reviewer for the suggestion to further clarify that the wound’s biochemical environment could trigger DNA damage. We note at the wound site damage is not as high in stem cells in transit through the migratory region distal to the wound site. The double immunostaining with anti-PAR [to detect DNA damage] and Anti-Tud1 [to detect damage in stem cells (Tud1+) and differentiated progenies (Tud1- cells)] allowed us to perform a controlled experiment where we can simultaneously analyse both migratory and non-migratory cells in the same environment. The non-migratory Tud1- differentiated cells that are present in the same environment and presumably would also get DNA damage as a potential bystander effect of the “wound’s biochemical environment” as the reviewers suggested. In this scenario, we only see increased DNA damage at 7dpa in TUD1+ stem cells in the migratory region compared to the shield and do not detect damage in TUD1- cells (Figure 2 D-F), once again implicating migration as the key variable. We have also confirmed this result by performing COMET assay using cells from the shielded region and migratory regions with and without migrating stem cells (Figure 2 G-K).

Can the authors assess whether migrating cells are more sensitive to oxidative stress when entering the wounded region, or whether DNA damage could be coupled to a specific phase of the cell cycle?

See comments above.

2. Can degradation due to apoptosis be compatible with the comet assay results?In Figure 3, can longer comet tails be a result of increased apoptosis after radiation exposure?

We thank the reviewer for bringing this up. The COMET results in control experiments after irradiation (Figure 1 and Figure 1—figure supplement 2 A-B) show reduction in COMET after IR exposure suggest COMET is indeed measuring damage that is repaired in the cell population. The extensive DNA fragmentation that occurs during apoptosis does not show comet-like structures hence comet tails are representative of DNA lesions and used for this purpose. Fragmented DNA in an apoptotic nuclei is very small (size of a nucleosome oligomer) and would generally disappear completely by diffusion in the gel during lysis and/or electrophoresis and are mostly seen as diffuse spheroids or like a halo around the nucleoid-head in single cell electrophoresis and do not show a structure that resembles comet-tail, rather they are referred to as hedgehogs (For interpretations of Comet assay results see, Lorenzo et al., Mutagenesis 2013, Collins 2004, Molecular Biotechnology). We have cited these papers in the manuscript to provide clear interpretation of our Comet assay data.

3. RNAi results are incompletely reported. According to the paragraph in the middle of page 4, data for knockdown of atr, atm, brca2, parp1, and parp2 under homeostatic conditions should be included in Figure S3. However, the data in this figure is phosphohistone-H3 staining instead of smedwi-1 staining, used in most other figures. In addition, the data in the figure are incomplete (not representing all genes tested). Rad51 is only mentioned, not shown. Approximate numbers of smedwi-1 cells in homeostatic conditions (either by qRT-PCR or in situ hybridization) need to be shown.

The number of *smedwi-1* cells in homeostatic conditions after knockdown of atr, atm, brca2, parp1, parp2, parp3, Rad51 and FancJ is included in the revised manuscript, please see Figure 1—figure supplement 3 I-J. The effect of RAD51 in stem cell maintenance is previously reported by Peiris et al., 2017, Development and we have added its effect in stem cell repopulation in the context of sub-lethal dose of IR in Figure 1H and its effect on animal survival in Figure 1—figure supplement 3 I-M.

Controls showing the efficacy of RNAi need to be included (either by qRT-PCR or in situ hybridization). The authors mention multiple times that they may be getting incomplete knockdown – why not just include some validation?

We have added the qRT-PCR validation of the genes tested in this revised manuscript, please see Figure 1—figure supplement 3. These show substantial knockdown after RNAi.

Furthermore, why is ATM RNAi excluded from Figure 1H (and methods?).

We have added the data of ATM RNAi after DDR knockdown and at homeostatic conditions, please See Figure 1H and Figure 1—figure supplement 3 I-M. The details regarding atm are also added in the methods section.

4. If the inability to repair DNA is solely responsible for a failure of stem cells to migrate to the wound, then cells should be seen dying en route to the wound, with elevated levels of PAR staining and increased comet tails. Is this the case?

We thank the reviewer for this question, to check if there is increased DNA damage during migration after RNAi of DNA repair factor, we need markers of upstream DNA damage signaling proteins (like phosphorylated-H2Ax and 53BP1) and live-cell imaging tools in planarians, which is beyond the scope and time frame of this manuscript. For example, *atr* and *atm* are the major kinase responsible for H2Ax phosphorylation in response to single or double stranded break induction. Hence PAR formation which is an ADP-ribosylation event after DNA break repair is initiated, will not be increased after knockdown of DNA repair factors. Moreover, in the absence of stem cells in the migratory region after RNAi, it will be difficult to interpret any DNA damage signals or Comet assay as there is no/few stem cells to check for by COMET, which does not discriminate stem cells from differentiated cells. The inability of *smedwi-1+* stem cells to successfully migrate after RNAi of DDR genes (Figure 4A) suggests a role for DDR in normal cell migration. Further work will be required to understand this process in greater detail.

The authors need to verify PAR staining specificity. An important control for these experiments would be a whole-animal stain of a shielded animal, which should show that the shielded area is devoid of PAR staining. The authors also need to indicate that quantification of PAR fluorescence was done in a standardized way, some indication of the number of animals analyzed, etc. Furthermore, the color selection makes it very difficult to see the staining in some images (e.g. Figure 2D). Higher magnification images and single-color images should be included.

We do indeed present data for cells in the shielded area as a control at all time points as the reviewers suggest (Figure 2D, E) and provide higher magnification images just 5 minutes after irradiation (Figure 2—figure supplement 2A).

We have kept individual PAR and Tud1 staining merged with Hoechst staining, providing clarity that Tud1 is peri-nuclear and PAR is nuclear, allowing independent measurement. The third panel represents a merged image of all three channels. We have remained consistent in representing all our PAR and Tud1 staining in this pattern (Please see Figure 1E, Figure 1—figure supplement 2 C).

The measurement of PAR is strictly nuclear, measuring only the signal overlapping with Hoechst. While a baseline signal of PAR is also detectable, the rise just 5 minutes after IR exposure is very clear, as its subsequent decrease over the next 24 hours. This control experiment (Figure 1E and Figure 1—figure supplement 2C) allows us to then measure DNA damage in migrating cells vs stationary cells with confidence. The PAR antibody has been widely used as a DNA damage marker in mammalian cell lines and previously in planarians (Shibata et al., Dev Cell 2017), and we characterize its specificity with controls that were previously absent.

We performed double immunostaining with anti-PAR (DNA damage marker) and an antibody to planarian Tudor-1 (that marks the perinuclear RNP granules (chromatoid bodies) in *smedwi-1+* stem cells (Figure 1—figure supplement 2 C)) and measured DNA damage in stem cells. Tudor-1 gives a very specific perinuclear staining (Solana et al., 2009), and hence aids in demonstrating PAR staining to be nuclear. All analyses are made by measuring total PAR fluorescence by outlining the nucleus based on Hoechst staining, and then Tud-1 perinuclear staining is used to differentiate between stem cells (Tud1+) and post mitotic progenies (Tud1-). Experiments are performed with 5 worms per condition. The images are from 10 µm paraffinized sections and Z stack to cover single-nuclear volumes and are used for analysis in Figure 1 E-G, Figure 2 D-F and Figure 1—figure supplement 2 D-F. Just showing a single channel makes it very difficult to judge if the staining is nuclear and hence in our manuscript, we have kept individual PAR or Tudor1 staining merged with the Hoechst channel, followed by a third panel with a merge of all 3 stains. We have also presented a representative image in Figure 2—figure supplement 2-A with Z-stack optical section and individual channels showing the specificity of the staining. The color selection for all the figures in the manuscript is based on CMYK format with Magenta and Yellow and Blue to make scientific figures accessible to readers with color-blindness, and we wish to stick to these colors.

How was the presence/absence of DDR components assessed?

Identification of orthologous genes across animal species were done using Inparanoid database (http://inparanoid.sbc.su.se/cgi-bin/index.cgi) and EggNOG (http://eggnogdb.embl.de/) databases. The phylogenetic tree is based on the Grohme et al., 2018. We also used reciprocal blastp result against the nr database and tblastn result against each sequence. The Planmine database was used for the identification of sequences of *S. mediterranea* and other flatworm species. We have added this information in the revised version of this manuscript as well as references to these databases.

5. Cite Guedelhoefer, Development, 2012, as this was the first publication to suggest that stem cell migration occurs in planarians.

We thank the reviewer for pointing out this citation which we have inadvertently missed in the manuscript. We have cited Guedelhoefer et al., Development, 2012 in the revised manuscript.

6. What is the relationship between the distance travelled by the stem cells and the amount of DNA damage?

We have analyzed this data and included in the revised manuscript. Please see Figure 2—figure supplement 2C.

References

1. P. Abnave, et al., Epithelial-mesenchymal transition transcription factors control pluripotent adult stem cell migration in vivo in planarians. Development 144, 3440–3453 (2017).2. M. A. Grohme, et al., The genome of *Schmidtea mediterranea* and the evolution of core cellular mechanisms. Nature 554, 56–61 (2018).3. T. H. Peiris, et al., Regional signals in the planarian body guide stem cell fate in the presence of genomic instability. Development 143, 1697–709 (2016).4. K. P. O’Brien, M. Remm, E. L. L. Sonnhammer, Inparanoid: a comprehensive database of eukaryotic orthologs. Nucleic Acids Res. 33, D476-80 (2005).5. J. Huerta-Cepas, et al., eggNOG 5.0: a hierarchical, functionally and phylogenetically annotated orthology resource based on 5090 organisms and 2502 viruses. Nucleic Acids Res. 47, D309–D314 (2019).6. A. Rozanski, et al., PlanMine 3.0-improvements to a mineable resource of flatworm biology and biodiversity. Nucleic Acids Res. 47, D812–D820 (2019).7. Y. Lorenzo, S. Costa, A. R. Collins, A. Azqueta, The comet assay, DNA damage, DNA repair and cytotoxicity: hedgehogs are not always dead. Mutagenesis 28, 427–32 (2013).8. J.-P. Baudoin, C. Alvarez, P. Gaspar, C. Métin, Nocodazole-induced changes in microtubule dynamics impair the morphology and directionality of migrating medial ganglionic eminence cells. Dev. Neurosci. 30, 132–43 (2008).9. A. Ganguly, H. Yang, R. Sharma, K. D. Patel, F. Cabral, The role of microtubules and their dynamics in cell migration. J. Biol. Chem. 287, 43359–69 (2012).10. N. Shibata, et al., Inheritance of a Nuclear PIWI from Pluripotent Stem Cells by Somatic Descendants Ensures Differentiation by Silencing Transposons in Planarian. Dev. Cell 37, 226–37 (2016).11. J. Solana, P. Lasko, R. Romero, Spoltud-1 is a chromatoid body component required for planarian long-term stem cell self-renewal. Dev. Biol. 328, 410–21 (2009).12. A. R. Collins, The comet assay for DNA damage and repair: principles, applications, and limitations. Mol. Biotechnol. 26, 249–61 (2004).